# Masked Autoencoders that Listen

**Po-Yao Huang**[1]   **Hu Xu**[1]   **Juncheng Li**[2]   **Alexei Baevski**[1]
**Michael Auli**[1]   **Wojciech Galuba**[1]   **Florian Metze**[1]   **Christoph Feichtenhofer**[1]

[1]Meta AI      [2]Carnegie Mellon University

## Abstract

This paper studies a simple extension of image-based Masked Autoencoders (MAE) [1] to self-supervised representation learning from audio spectrograms. Following the Transformer encoder-decoder design in MAE, our Audio-MAE first encodes audio spectrogram patches with a high masking ratio, feeding only the non-masked tokens through encoder layers. The decoder then re-orders and decodes the encoded context padded with mask tokens, in order to reconstruct the input spectrogram. We find it beneficial to incorporate local window attention in the decoder, as audio spectrograms are highly correlated in local time and frequency bands. We then fine-tune the encoder with a lower masking ratio on target datasets. Empirically, Audio-MAE sets new state-of-the-art performance on six audio and speech classification tasks, outperforming other recent models that use external supervised pre-training. Our code and models is available at https://github.com/facebookresearch/AudioMAE.

## 1   Introduction

Transformers [2] and self-supervised learning [3, 4, 5, 6, 7, 1] are dominating computer vision (CV) and natural language processing (NLP) research. The revolution firstly started in NLP with the invention of the Transformer architecture and self-attention [8]. Masked autoencoding with BERT [3] set a new state-of-the-art on various NLP tasks by self-supervised pre-training on large-scale language corpus. Similarly in the CV community, Vision Transformers (ViT) [9] have become popular for CV tasks, and, for self-supervised image representation learning, Masked Autoencoders (MAE) [1] have brought the CV community closer to the success of BERT in NLP. In addition to the existing masked autoencoders that can read (BERT) or see (MAE), in this work we study those that can *listen*.

Transformer-based models have recently refreshed leaderboards for audio understanding tasks. For example, AST [10] and MBT [11] improved the audio classification performance on the AudioSet [12], Event Sound Classification [13], etc. The key technique behind this is initialization of audio model weights with ImageNet pre-trained supervised models (*e.g.*, DeiT [14]) by deflating patch embeddings and interpolating positional embeddings for encoding audio spectrograms. However, exploiting ImageNet pre-trained models could be sub-optimal. Unlike initializing video models with weights from image models (*e.g.*, the initial weights of I3D [15] or 3D-ResNets [16] are inflated from ImageNet pre-trained image models), there are clear and notable discrepancies between spectrograms representing audio content and natural images. It remains unclear why such heterogeneous image-to-audio transfer is useful beyond arguably similar low-level semantics such as shapes of spectrograms and shapes of visual objects. Further, any label bias would inevitably be transferred to audio models.

Addressing these concerns, self-supervised audio representation learning has recently attracted much research attention. Based on BEiT [17] that learns to reconstruct image patches or learnt patch tokens, SS-AST [18] extends to the audio domain and exploits spectrograms (akin to 1-channel 2D images) and use both contrastive and reconstruction objective as self-supervision. Without using any labels, the key enabler to effective self-supervised representation learning is large-scale pre-training data. In this work we use AudioSet [12] for pre-training, a common dataset containing ~2 million audio recordings. Performing large-scale training with Transformer architectures is challenging as self-attention in Transformers has quadratic complexity w.r.t. the length of input sequence.

36th Conference on Neural Information Processing Systems (NeurIPS 2022).

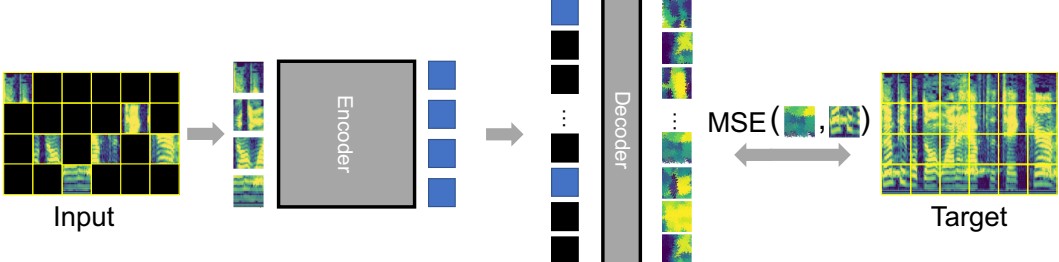

Figure 1: **Audio-MAE for audio self-supervised learning**. An audio recording is first transformed into a spectrogram and split into patches. We embed patches and mask out a large subset (80%). An encoder then operates on the visible (20%) patch embeddings. Finally, a decoder processes the order-restored embeddings and mask tokens to reconstruct the input. Audio-MAE is minimizing the mean square error (MSE) on the masked portion of the reconstruction and the input spectrogram.

This computational burden has been addressed in different ways. A popular approach is to reduce the sequence length in self-attention. Various ViT-based architectures have been developed to alleviate such issues for image and video understanding. For example, Swin-Transformer [19] only performs local attention within windows that shift across layers. MViT [20] employs pooling attention to construct a hierarchy of Transformers where sequence lengths are downsampled. For self-supervised learning, MAE [1] efficiently encodes only a small portion (25%) of visual patches while the majority of patches is discarded. The simplicity and scalability in MAE make it a promising framework for large-scale self-supervised learning.

In this work, we study MAE for sound recognition and the unique challenges of the audio domain. We present Audio-MAE (Fig. 1) as unified and scalable framework for learning self-supervised audio representations. Similar to MAE, it is composed of a pair of a Transformer encoder and decoder. Sound is first transformed and embedded into spectrogram patches. Before feeding them into the Transformer encoder, we mask and discard the majority and only feed a small number of non-masked embeddings into the encoder for efficient encoding. After padding encoded patches with learnable embeddings to represent masked patches, it then restores the order of these patches in frequency and time and propagates them through a Transformer decoder to reconstruct the audio spectrogram.

Different from image patches, spectrogram patches are comparably local-correlated. For example, formants, the vocal tract resonances, are typically grouped and continuous locally in the spectrogram. The location in frequency and time embeds essential information that determines the semantics of a spectrogram patch and how it sounds like. To this end, we further investigate using localized attention and a hybrid architecture in the Transformer decoder to properly decode for reconstruction. This simple-yet-effective upgrade leads to improved performance for Audio-MAE.

Similar to MAE for images, we minimize the patch-normalized mean square error. At the fine-tuning stage, we discard the decoder and fine-tune the encoder with patch-masking. Empirically, Audio-MAE sets a new state-of-the-art performance on six audio and speech classification tasks. It is the first audio-only self-supervised model that achieves state-of-the-art mAP on AudioSet-2M, outperforming other recent models with external supervision. We further provide the visualization and audible examples to qualitatively demonstrate the effectiveness of the Audio-MAE decoder.

## 2 Related Work

**Visual masked pre-training.** Masked/Denoising autoencoders [21, 22, 3] are a general representation learning methodology by reconstructing source from masked or corrupted inputs. In CV, visual masked pre-training has made recent progress [23, 24, 1, 20]. Based on ViT [9] that applies Transformers to image patches, BEiT [17] and MAE [1] present masked image modeling frameworks. BEiT [17] learns to predict discrete visual tokens generated by VAE [25] in masked patches. MAE [1] reduces sequence length by masking a large portion of image patches randomly and encoding only non-masked ones for reconstruction of pixel color information. MaskFeat [20] studies features for masked pre-training and finds that Histograms of Oriented Gradients (HoG) [26], which are in turn related to spectrogram features, perform strongly for image and video classification models. Our work extends the MAE framework for representation learning with audio spectrograms.

**Out-of-domain pre-training for audio.** Transferring ImageNet supervised pre-trained ViT [9] or ResNet [27] has become a popular practice for audio models [10, 28, 11, 29, 30, 31]. After pre-training, these models operate over audio spectrograms by deflating from 3-channels (RGB) into 1-channel (spectrogram) in the pre-trained patch embedding in ViT and employing the rest of the transformer blocks on top. For example, HTS-AT [29] encodes spectrograms with hierarchical Transformer initialized from the Swin Transformer [19]. MBT [11] uses ImageNet-21K pre-trained ViT; AST [10] and PaSST [28] employ DeiT [14] as the Transformer backbone. Without using out-of-domain (non-audio) data, the proposed Audio-MAE focuses on audio-only self-supervised pre-training from scratch.

**In-domain pre-training for audio.** Existing in-domain (*i.e.*, audio-only) self-supervised methods can be broadly categorized by the input signal type (*e.g.*, raw waveform [32, 33, 34], frame-level features [35, 36, 37], or spectrogram patches [18, 38]); and the objective used for self-supervision (*e.g.*, contrastive [39, 33, 40, 41, 35] or prediction/reconstruction [18, 34, 37, 36]). For example, wav2vec 2.0 [33] takes raw waveform as inputs and exploits contrastive learning to discriminate contextualized representations in different time segments. Mockingjay [42] proposed a masked acoustic model pretext task to reconstruct frame-level Mel-features of masked time frames. SS-AST [18] is the closest work to Audio-MAE and is our main benchmark. Inspired by the success of BERT [3], SS-AST proposed a self-supervised learning method which operates over spectrogram patches and employs joint contrastive and reconstructive objectives on masked patches. These previous methods generate audio representations by encoding full-view of both masked and non-masked time or spectrogram segments for self-supervised pre-training. In contrast, Audio-MAE encodes only the non-masked spectrogram patches.

Our work is done independently and concurrently with [38, 43, 44] related methods. We also compare our model to these concurrent works in the experiments and showcase the superiority of Audio-MAE.

## 3 Audio Masked Autoencoders (Audio-MAE)

Audio-MAE is a conceptually simple extension of MAE to learn self-supervised representations from audio spectrograms. Fig. 1 depicts an overview. The details of each component are as follows.

**Spectrogram Patch Embeddings**. Following [10, 18], we transform audio recordings into Mel-spectrograms and divide them into non-overlapped regular grid patches. These patches are then flattened and embedded by a linear projection. Similar to MAE [1], we add fixed sinusoidal positional embeddings to the embedded patches.

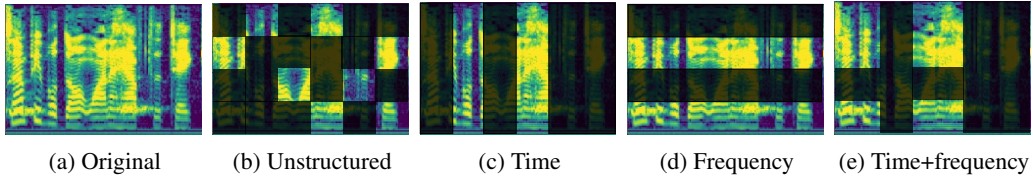

(a) Original  (b) Unstructured  (c) Time  (d) Frequency  (e) Time+frequency

Figure 2: Audio-MAE's masking strategies on Mel-spectrograms.

**Masking Strategies**. Audio-MAE masks out a large subset of spectrogram patches. As a spectrogram can be viewed as a 2D representation of time and frequency components of a sound, it is reasonable to explore treating time and frequency differently during masking. In this work, we investigate both the *unstructured* (*i.e.*, random masking without any prior) and *structured* (*i.e.*, randomly masking a portion of time, frequency, or time+frequency of a spectrogram) in the pre-training and fine-tuning phase. Illustrative examples are shown in Fig. 2. We show masked regions with dark overlay.

The masking mechanism, as introduced in MAE [1], is the key ingredient for efficient self-supervised learning. For a input patch sequence, this can be regarded as a Bernoulli process where each patch is masked/dropped with probability $p$ (masking ratio). Masking reduces input patch sequence length and encourages learning global, contextualized representations from limited "visible" patches. We observe that akin to images, a large masking rate (80% in our experiments for spectrogram patches, which is similar to 75% in MAE for images) is feasible for learning self-supervised audio representations. Unlike BERT [3] that uses 15% masking rate for self-supervised learning in NLP, most of the

tokens/patches can be discarded for spectrograms as well as images due to high redundancy in these modalities. Beyond self-supervised pre-training, we further explore the effectiveness of masking in the supervised fine-tuning stage. Empirically, we found unstructured (random) masking at a higher ratio for pre-training and structured (time+frequency masking) at a lower ratio for fine-tuning provide best accuracy (ablations are in §4.4).

**Encoder**. Audio-MAE uses a stack of standard Transformers [2] as its encoder. The encoder only processes (20%) non-masked patches to reduce computation overhead which is quadratic to the input sequence length. We use the 12-layer ViT-Base (ViT-B) [9] Transformer as our default.

**Decoder with Local Attention**. The decoder is also composed of standard Transformer blocks. The encoded patches from the encoder are padded with trainable masked tokens. After restoring the original time-frequency order in the audio spectrogram, we add the decoder's (fixed sinusoidal) positional embeddings and feed the restored sequence into the decoder. At the top of the decoder stack, we add a linear head to predict and reconstruct the input spectrogram.

To address the unique characteristics of audio spectrograms, our work investigates an enhancement to the vanilla MAE decoder. Image-based MAE uses *global self-attention* in the Transformer decoder which is appropriate for visual context, because visual objects are typically invariant under translation or scaling, and their exact position may not affect the semantics of an image. In contrast, the position, scale, and translation of spectrogram features however *directly affects* the sound or semantics of an audio recording. Consequently, global self-attention is sub-optimal for spectrograms if the time-frequency components is predominantly local. For instance, we would have better success to use the harmonics (*e.g.*, Fig. 2a) in lower bands of a vowel to predict the spectrogram patch vertically in a higher frequency band rather than horizontally in the time domain. Similarly, a frictional sound of a consonant likely only correlates to other part of the consonant, and is without dependency to other silence segments in the audio recording. Compared to images, the spectrogram patches are more similar to speech or text tokens where its order and position is more relevant.

To address the nature of audio spectrograms, in addition to using Transformers with global self-attention as in vanilla MAE, we incorporate the *local attention mechanism* which groups and separates the spectrogram patches in to local windows in self-attention for decoding. We investigate two types of local attention: (1) Shifted window location: Inspired by the shifted-window in Swin Transformers [19], we shift window attention by 50% between consecutive Transformer decoder layers. For padding the margin when shifting, we cyclically shift the spectrogram to the top-left direction. Fig. 3 illustrates the localized decoder attention by shifted windows. (2) Hybrid window attention (global+local attention): Inspired by [45], to add better cross-window connections, we design a simple hybrid (global+local) attention that computes local attention within a window in all but the last few top layers. In this way, the input feature maps for the final reconstruction layer also contain global information. For simplicity, we use *no* pooling or hierarchical structure. Decoders with different attention types are compared in §4.4.

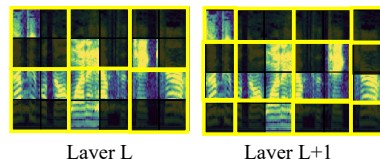

Layer L      Layer L+1

Figure 3: Decoder's local attention and shifted window (right).

**Objective**. The Audio-MAE decoder learns to reconstruct the input spectrogram by predicting the values in the spectrogram patches or their per-patch normalized ones. The objective is the mean squared error (MSE) between the prediction and the input spectrogram, averaged over unknown patches. Empirically we found employing the reconstruction loss alone is sufficient while including additional contrastive objectives (*e.g.*, InfoNCE loss [46]) does not improve Audio-MAE.

**Fine-tuning for Downstream Tasks**. In the fine-tuning stage, we only keep and fine-tune the Audio-MAE encoder and discard the decoder. Different from the original MAE, and inspired by [47, 28], we also explore to employ masking in the fine-tuning stage to remove a portion of patches to further regularize learning from a limited view of spectrogram inputs, which, as a side effect, also reduces computation during fine-tuning. Compared to SpecAug [48] which takes full-length input with the masked portion set to zero as data augmentation, Audio-MAE sees only a subset of real-valued input patches without the nullified ones. Audio-MAE then encodes these non-masked patches and applies an average pooling layer followed by a linear layer on top for fine-tuning in classification tasks.

# 4 Experiments

We perform an extensive evaluation on six tasks, including audio classification on AudioSet (AS-2M, AS-20K) and Environmental Sound Classification (ESC-50), and speech classification on Speech Commands (SPC-1 and SPC-2) and VoxCeleb (SID). We use AudioSet for ablation studies.

## 4.1 Datasets and Tasks

**AudioSet** [12] (AS-2M, AS-20K) contains ~2 million 10-second YouTube clips for audio classification. 527 types of audio events are weakly annotated [49, 50, 51] for each clip. There could be multiple events in a clip. The *full* training set has 2 subsets: A class-wise *balanced* (22,176 clips) and an *unbalanced* (2,042,985 clips) set. The *eval* set has 20,383 clips. We downloaded and processed around 1.96M unbalanced training, 21K balanced training, and 19K evaluation clips.

For the AS-2M experiments, we use the union of unbalanced and balanced training audio for pre-training and fine-tuning. For the AS-20K experiments, we use AS-2M for pre-training and the 20K balanced set for fine-tuning. We report the testing mAP on the 19K *eval* set used by AST [10].

**Environmental Sound Classification** (ESC-50) [13] is an audio classification dataset consists of 2,000 5-second environmental sound recordings. There are 50 classes in ESC. We report accuracy under 5-fold cross-validation with the same split used by [10].

**Speech Commands** (SPC-2, SPC-1) [52] are two keyword spotting tasks. In SPC-2, there are 35 speech commands. The training/validation/testing set has 84,843/9,981/11,005 1-second recordings, respectively. In SPC-1, there are 10 classes of keywords, 1 silence class, and 1 unknown class that includes all the other 20 common speech commands. We use the data and split provided in the SUPERB [53] benchmark to report the testing accuracy.

**VoxCeleb** (SID) [54] contains 150K utterances from 1,251 speakers. The speaker identification task (SID) is to classify the utterances to identify its original speaker. We use the V1 standard train (138,361), validation (6,904), testing (8,251) sets and report the testing accuracy.

## 4.2 Implementation Details

We use a vanilla 12-layer ViT-B by default as the Transformer encoder. For the decoder, we use a 16-layer Transformer with shifted local attention. We investigate the vanilla (global attention) and hybrid (global+local attention) decoder variants (see Table. 1c).

Following [10, 11], we transform raw waveform (pre-processed as mono channel under 16,000 sampling rate) into 128 Kaldi [55]-compatible Mel-frequency bands with a 25ms Hanning window that shifts every 10 ms. For a 10-second recording in AudioSet, the resulting spectrogram is of $1 \times 1024 \times 128$ dimension.

For patch embedding, we use convolutional kernels with $(16, 16)$ size and stride in time and frequency (thus, patches are non-overlapping) to avoid short-cuts via overlap in self-supervision (though, at high masking ratios such short-cuts are less severe). By default, we use a masking ratio of $0.8$ with (unstructured) random masking for pre-training. During fine-tuning, we employ a lower masking ratio ($0.3$ in time and $0.3$ in frequency). Ablations on these design choices are given in §4.4.

## 4.3 Pre-training and Fine-tuning

We use AudioSet-2M for pre-training and randomly iterate over all audio recordings. We train for 32 epochs with a batch size of 512 and a 0.0002 learning rate. We distribute the training load over 64 V100 GPUs and the total training time is ~36 hours. For each audio, we randomly sample the starting time, cyclically extract 10-second audio, and randomly jitter its magnitude by up to $\pm$ 6dB. We use only natural audio spectrograms and apply *no* augmentations (*e.g.*, [48, 56, 57]) as we do not find these strong augmentations helpful in the pre-training phase.

In the fine-tuning phase, we remove the decoder and only fine-tune the encoder. For the supervised fine-tuning on AudioSet-2M, since the size of training samples are uneven across classes (unbalanced), we follow the common practice of using a weighted sampling to balance the classes during training. In each epoch, we sample 200K instances (~10% of AudioSet-2M) without replacement. We fine-tune

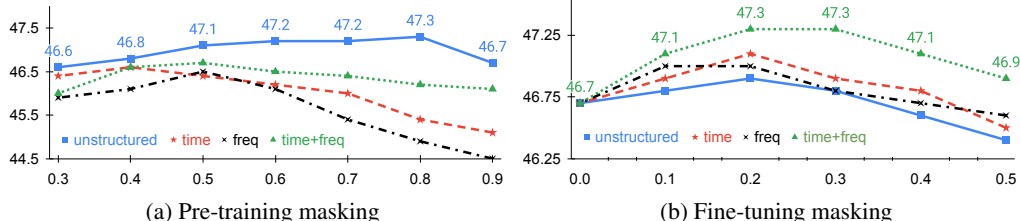

| (a) Pre-training masking | (b) Fine-tuning masking |

Figure 4: **Masking strategy**. For pre-training, a *higher* ratio and *unstructured* masking (random) is preferred. For fine-tuning, a *lower* ratio and *structured* masking (time+frequency) is better. The y-axes are mAP on AS-2M and the x-axes are masking ratio. This ablation format follows [1].

for 100 epochs, which aggregate to ~10 full epochs of AudioSet-2M. The probability of sampling an instance is inversely proportional to the dataset-wise occurrences of its classes. Fine-tuning on 64 GPUs takes ~12 hours. For the smaller balanced AudioSet-20K, we fine-tune on 4 GPUs for 60 epochs without weighted sampling. Please see Supplementary for the details on other datasets.

## 4.4 Ablations and Model Properties

**Masking Strategies in Pre-training and Fine-tuning.** In Fig. 4, we compare different pre-training and fine-tuning masking strategies for Audio-MAE. First, in Fig. 4a we explore the *pre-training masking ratio*. We observe, similar as in MAE for images [1], that a high pre-training masking ratio (80% in our case) is optimal for audio spectrograms. This is due to the fact that both audio spectrograms and images are continuous signals with significant redundancy. Further, we find the unstructured random masking works the best for self-supervised pre-training over more structured masking (*e.g.*, time+frequency).

Unlike MAE for images, there are clear performance differences among masking strategies when pre-training with audio spectrograms. Comparing Audio-MAE reconstructions between Fig. 6a to 6e and 6d to 6h, under the same masking ratio, we observe the unstructured random masking is comparably easier than structured masking (*i.e.*, time and/or frequency) as the model can guess the missing component by extrapolating nearby context (*e.g.*, formants in vowels and frictional sounds in consonants around). We also observe that for higher masking ratios, the structured masking alternatives drop in performance, presumably because the task becomes too difficult while random masking improves steadily up to 80%. This result show that designing a pretext task with *proper hardness* is important for effective self-supervised learning of audio representations. We therefore use random masking with ratio of 80% as our default for pre-training.

Fig. 4b studies the effect of masking during the *fine-tuning* phase. We see that in this case, it is more beneficial to use structured masking: time+frequency performs better than time- or frequency-based masking, and these perform better than unstructured masking. Overall, we see that the optimal masking ratios are *lower* than for pre-training and we use 0.3 as our default in the fine-tuning phase.

In general, we observe that for task-agnostic pre-training, unstructured masking with a higher ratio is preferred. While in task-specific fine-tuning, structured masking with lower ratios performs better.

**Impact of Patch Size and Stride.** We compare the performance of Audio-MAE trained with different patch sizes and strides in Table 1a. A non-zero overlap (*i.e.*, stride < patch size) between patches will increase the number of patches and quadratically increase computation in floating point operations (FLOPs), as reported in the table. Most prior works follow AST [10] to use overlapped patches (patch = 16 and stride = 10) to boost end task performance. As shown in Table 1a, we do not observe a performance improvement using overlapped patches for Audio-MAE (both 47.3 mAP), presumably because due to overlap, the patch embedding can leak information into the masked patches. The non-overlapped 16×16 patches achieve a good balance between computation and performance. By default, we use this setup in our experiments.

**Encoder.** We investigate the design choices of encoder and decoder architectures in Audio-MAE. Table 1b shows the trade-off between encoder model size and performance. As expected, larger models achieve better performance, at a cost of computation and memory. The accuracy gain of ViT-L over ViT-B/S is more significant on the smaller and balanced AS-20K. For ViT-S, the performance

| Patch size, stride | Seq shape | FLOPs | mAP |
|---|---|---|---|
| (16,16), (16,16) | 64×8 | 48.6 | **47.3** |
| (16,16), (10,10) | 101×12 | 130.5 | **47.3** |
| (32,16), (16,16) | 63×8 | 47.8 | 46.6 |
| (16,32), (16,16) | 64×7 | 42.1 | 46.8 |

(a) **Patch size and stride**

| Backbone | #Params | AS-20K | AS-2M |
|---|---|---|---|
| ViT-S | 22M | 32.1 | 45.0 |
| ViT-B | 86M | 37.1 | 47.3 |
| ViT-L | 304M | **37.6** | **47.4** |

(b) **Model size (encoder)**

| Attention type | AS-20K | AS-2M | ESC-50 | SID |
|---|---|---|---|---|
| Global$^{(8)}$ (vanilla) | 36.6 | 46.8 | 93.6 | 94.1 |
| Local$^{(16)}$ (shifted) | **37.1** | **47.3** | **94.1** | 94.8 |
| Hwin (local$^{(8)}$+ global$^{(4)}$) | 36.8 | **47.3** | 93.8 | **95.0** |

(c) **Decoder attention comparison**. Attn type$^{(depth)}$

| Depth | mAP |
|---|---|
| 2 | 46.8 |
| 8 | 47.2 |
| 16 | **47.3** |

(d) **Decoder depth**

| Width | mAP |
|---|---|
| 256 | 46.9 |
| 512 | **47.3** |
| 768 | 47.3 |

(e) **Decoder width**

| % of AS-2M | mAP |
|---|---|
| 1% (AS-20K) | 39.4 |
| 1% (AS-2M) | 39.6 |
| 10% | 42.6 |
| 50% | 46.4 |
| 100% | **47.3** |

(f) **Pre-training size**

| epoch | mAP |
|---|---|
| 8 | 46.5 |
| 16 | 46.8 |
| 24 | 47.2 |
| 32 | **47.3** |
| 40 | 47.3 |

(g) **Pre-training epoch**

| scenario | IN-SSL | IN-SL | AS-SSL | AS-20K | AS-2M |
|---|---|---|---|---|---|
| (1) | | | ✓ | **37.1** (-0.0) | **47.3** (-0.0) |
| (2) | ✓ | | | 32.1 (-5.0) | 45.4 (-1.9) |
| (2) | ✓ | ✓ | | 32.5 (-4.6) | 45.9 (-1.4) |
| (3) | ✓ | | ✓ | 36.9 (-0.2) | 47.1 (-0.2) |
| (3) | ✓ | ✓ | ✓ | 36.2 (-0.9) | 46.9 (-0.4) |

(h) **External ImageNet (IN) pre-training**. SSL: w/ self-supervised MAE. SL: w/ supervised (fine-tuned) MAE.

Table 1: **Ablation studies on AS-2M**. The gray entries are the default Audio-MAE setup (ViT-B encoder, decoder with shifted local attention, pre-trained for 32 epochs). Table format follows [1].

gap to ViT-B can be significantly closed (5.0 → 2.3 mAP) when fine-tuning with more in-domain data (AS-20K → AS-2M).

**Decoder.** Table 1c compares decoder attention types in Audio-MAE. Note that decoders are discarded after pre-training and only the equal-sized ViT-B encoders are fine-tuned for the end task. Our results show that *local attention* with shifted window achieves the best performance. Combining local and global attention (*i.e.*, hybrid attention, Hwin) also improves vanilla global self-attention. Fig. 5 shows the qualitative reconstruction comparison. In the spectrogram of vowels, the decoder with local attention reconstructs better harmonics and recovers more context in the spectrogram. Similar phenomena are observed in the frictional sound in the middle consonant.

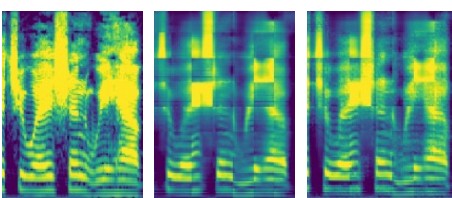

Ground-Truth  w/ global attention  w/ local attention

Figure 5: Decoder reconstruction comparison.

Table 1d ablates the impact of decoder depth on mAP. A deeper 16-layer decoder achieves better performance against its shallower variants. Note that our decoder uses local window attention by default where only a fraction of tokens (4×4 local windows *vs.* 64×8 with global attention) are attended. For global attention we find 8-layer decoders to perform better than 16-layer. Table 1e compares decoder width (embedding dimension). A 512-dimension decoder achieves a good trade-off between computation and performance as a wider one is not better.

**Pre-training Data and Setup.** Table 1f summarizes the impact of pre-training dataset size. Overall the model performance is monotonically increasing when using more data for pre-training. Comparing the performance of using 1% well-annotated AS-20K balanced data to using randomly sampled 20K unbalanced data for pre-training, the similar mAPs (39.4 vs 39.6) suggest that the *distribution* of data classes (balanced vs. unbalanced) is *less* important for pre-training. Meanwhile, as shown in Table 1g, training for longer is beneficial yet the performance saturates after the 24-*th* epoch.

**Out-of-domain Pre-training on ImageNet.** Initializing audio models from ImageNet pre-trained weights has become popular for audio classification. However, as there are significant discrepancies between image and audio modalities, it is questionable if out-of-domain pre-training benefits audio representation learning. In Table 1h we design 3 scenarios to investigate this for Audio-MAE: (1)

| Model | Backbone | PT-Data | AS-20K | AS-2M | ESC-50 | SPC-2 | SPC-1 | SID |
|---|---|---|---|---|---|---|---|---|
| **No pre-training** | | | | | | | | |
| ERANN [58] | CNN | - | - | 45.0 | 89.2 | - | - | - |
| PANN [59] | CNN | - | 27.8 | 43.1 | 83.3 | 61.8 | - | - |
| **In-domain self-supervised pre-training** | | | | | | | | |
| wav2vec 2.0 [33] | Transformer | LS | - | - | - | - | 96.2* | 75.2* |
| HuBERT [35] | Transformer | LS | - | - | - | - | 96.3* | 81.4* |
| Conformer [37] | Conformer | AS | - | 41.1 | 88.0 | - | - | - |
| SS-AST [18] | ViT-B | AS+LS | 31.0 | - | 88.8 | 98.0 | 96.0 | 64.3 |
| *Concurrent MAE-based works* | | | | | | | | |
| MaskSpec [43] | ViT-B | AS | 32.3 | 47.1 | 89.6 | 97.7 | - | - |
| MAE-AST [38] | ViT-B | AS+LS | 30.6 | - | 90.0 | 97.9 | 95.8 | 63.3 |
| **Audio-MAE** (global) | ViT-B | AS | 36.6±.11 | 46.8±.06 | 93.6±.11 | **98.3**±.06 | **97.6**±.06 | 94.1±.06 |
| **Audio-MAE** (local) | ViT-B | AS | **37.0**±.11 | **47.3**±.11 | **94.1**±.10 | **98.3**±.06 | 96.9±.00 | **94.8**±.11 |
| **Out-of-domain supervised pre-training** | | | | | | | | |
| PSLA [30] | EffNet [60] | IN | 31.9 | 44.4 | - | 96.3 | - | - |
| AST [10] | DeiT-B | IN | 34.7 | 45.9 | 88.7 | 98.1 | 95.5 | 41.1 |
| MBT [11] | ViT-B | IN-21K | 31.3 | 44.3 | - | - | - | - |
| HTS-AT [29] | Swin-B | IN | - | 47.1 | 97.0† | 98.0 | - | - |
| PaSST [28] | DeiT-B | IN | - | 47.1 | 96.8† | - | - | - |

Table 2: **Comparison with other state-of-the-art models** on audio and speech classification tasks. Metrics are mAP for AS and accuracy (%) for ESC/SPC/SID. For pre-training (PT) dataset, AS:AudioSet, LS:LibriSpeech, and IN:ImageNet. †: Fine-tuning results with additional supervised training on AS-2M. We gray-out models pre-trained with external non-audio datasets (*e.g.*, ImageNet). Best single models in AS-2M are compared (no ensembles). *: linear evaluation results from [53].

Audio-only pre-training (AS-SSL) from scratch. We consider this the ideal schema for learning audio representations as it is a simple and clean setup that prevents uncontrollable bias transfer from other modalities. (2) Directly using self-supervised ImageNet MAE models (IN-SSL) and its fine-tuned variant (IN-SL). (3) Audio-MAE self-supervised pre-training on top of these ImageNet weights.

The results show that (1) from-scratch *audio-only* pre-training is the best. For scenarios (2) and (3), we observe that ImageNet pre-training alone (2) is not sufficient (especially when the downstream data is smaller, AS-20K), and, in self-supervised pre-training on AudioSet, ImageNet initialization (3) does not help but degrades accuracy. Also in (3), supervised ImageNet pre-training (IN-SL) seems harmful. Consequently, the result suggests that out-of-domain pre-training (*i.e.*, ImageNet) is not helpful for Audio-MAE, possibly due to domain shift.

## 4.5 Comparison with the State-of-the-art

Table 2 compares Audio-MAE (with 3-run error bars) to prior state-of-the-art. We categorize the comparison into 3 groups. For fair comparison, our main benchmark is the models in the middle group with self-supervised pre-training on in-domain (audio) datasets (AudioSet and LibriSpeech). For reference we also list other models without pre-training (the top group) and other models with supervised pre-training on out-of-domain ImageNet (the bottom group), where the latter contains previous best systems on the datasets.

Pre-trained on AudioSet, Audio-MAE achieves the best performance across all tasks compared to other models with in-domain self-supervised pre-training. On AudioSet-20K, its 37.1 mAP significantly outperforms all other approaches including concurrent works and other models with out-of-domain pre-training. On AudioSet-2M and ESC-50, our method also outperforms Conformer [37] and SS-AST [18]. Notably, unlike SS-AST and concurrent MAE-AST [38], which trained with additional 1,000 hours of speech in Librispeech, we use only AudioSet for pre-training.

In the bottom group of Table 2, Audio-MAE also outperforms previous state-of-the-art models with ImageNet supervised pre-training. Note that the proposed Audio-MAE does not rely on any out-of-domain data and labels, nor using knowledge distillation (*e.g.*, DeiT) from additional CNN-based models. Also, compared to HTS-AT [29] and PaSST [28], Audio-MAE is trained with audio under 16K sampling rate. As experimented in [59], there could be up to 0.4 potential mAP improvement for Audio-MAE if audio with 32K sampling rate are available.

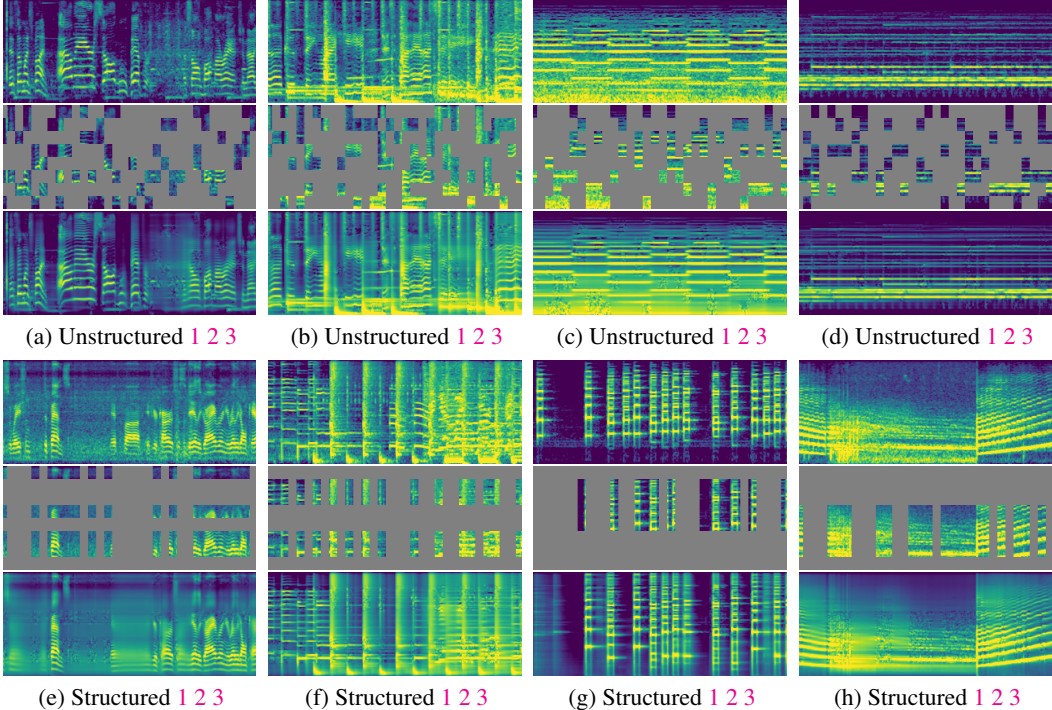

(a) Unstructured 1 2 3     (b) Unstructured 1 2 3     (c) Unstructured 1 2 3     (d) Unstructured 1 2 3

(e) Structured 1 2 3     (f) Structured 1 2 3     (g) Structured 1 2 3     (h) Structured 1 2 3

Figure 6: **Spectrogram reconstruction visualizations on the AudioSet *eval* set**. Column-wise type: speech, music, event, others. Masking type: (a-d) unstructured (random); (e-h) structured (time+frequency). Masking Ratio: 70%. In each group, we show the original spectrogram (1, top), masked input (2, middle), and MAE output (3, bottom). The spectrogram size is 1024×128; patch size is 16×16. Each sample has 64×8=512 patches with 154 (70% masked) patches being visible to Audio-MAE. Please click (1 2 3) for audible *.wav*s. More audible examples are in Supplementary.

For the speech tasks (SPC-1, SPC-2, and SID), Audio-MAE outperforms other models without pre-training (ERANN [58], PANN [59]), supervised (AST) and self-supervised models (SS-AST, MAE-AST). We further list other works (marked with *) to include the latest results introduced in the SUPERB [53] benchmark. But note that these results are not strictly comparable since SUPERB employs linear evaluation where the underlying pre-trained models are not end-to-end fine-tuned.

In summary, with audio-only from-scratch pre-training on AudioSet, our Audio-MAE performs well for both the audio and speech classification tasks.

### 4.6 Visualization and Audible Examples by Audio-MAE Decoder

For better visualization, we follow MAE [1] to use MSE over non-normalized spectrograms as the self-supervised objective. We use ViT-L as the Audio-MAE encoder for visualization. Fig. 6 illustrates the reconstruction results sampled from the AudioSet-2M *eval* set. We further reconstruct *.wav*s using the Griffin-Lim [61] algorithm, audible under the anonymous links (accessible in respective 1 2 3).

As can be seen and heard, for various masking strategies and different sounds, our Audio-MAE generates reasonable reconstruction. It works well for noisy event sounds (*e.g.*, the reconstructed siren in Fig. 6c-3), as well as speech and music (*e.g.*, the reconstructed singing in Fig. 6b-3). Notably, unlike visual contents that are typically scale/translation/position invariant [19], absolute positions and arrangement of spectrogram components are critical for humans to understand sound [62]. For example, shifting a pitch will make an audio sounds completely different. Also, phoneme sequences in time are important cues for speech understanding. Consequently, unstructured masking produces better aligned outputs that are closer to the ground-truth (top row in each subfigure) as the model can make better predictions based on nearby spectrogram patches; while structured masking is harder (less accurate or with words missing), especially when masking is performed over the time axis. A failure example (missing words) is the reconstructed speech in Fig. 6e-3.

# 5   Conclusion

We have explored a simple extension of MAE [1] to audio data. Our Audio-MAE learns to reconstruct masked spectrogram patches from audio recordings and achieves state-of-the-art performance on six audio and speech classification tasks. We have drawn four interesting observations: First, a simple MAE approach works surprisingly well for audio spectrograms. Second, we find that it is possible to learn stronger representations with local self-attention in the decoder. Third, we show that masking can be applied to both pre-training and fine-tuning, improving accuracy and reducing training computation. The optimal strategy depends on the nature of the data (audio, image, *etc.*) and the learning type (self-/supervised). Fourth, the best performance can be achieved by pre-training and fine-tuning under the same modality, without reliance on cross-modality transfer learning. In future work, we aim to explore multimodal self-supervised learning with a joint audio-visual MAE approach as these domains share natural correspondences in video data.

**Acknowledgements.** We thank Kaiming He and Luke Zettlemoyer for their feedback and discussions.

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
