# OpenReview forum: "Masked Autoencoders that Listen"
_NeurIPS.cc/2022/Conference — NeurIPS 2022 Accept_

### Official Review · Reviewer_NJ9B · 2022-07-11

**Rating:** 4
**Confidence:** 4
**Soundness:** 3 good
**Presentation:** 3 good
**Contribution:** 1 poor

**Summary:**

This paper presents the application of Masked Autoencoders (MAE) to audio spectrograms in a self-supervised framework. The overall idea is to rely on masking to widely corrupt the input data and train a Transformer AE to fill the blanked part of the spectrogram, leading to a form of inpainting self-supervised task. The authors study several different masking strategies in the context of spectrogram, and also analyse the impact of local or global attention in the decoder. The quality of the proposed method is assessed in a downstream speech and classification task.

**Questions:**

- My major question is on what you call a "unstructured" strategy, if we refer to Figure 2, I do not see any conceptual difference between (b) and (d), given that the "unstructured" is simply also a selection of time-frequency patches with a given size ? I think a truly unstructured approach would rather look like a Bernoulli dropout over the whole spectrogram ?
- I quite disagree with the overall premise (l.57) that "unlike image patches, spectrogram patches are mostly locally correlated" ... and I even think that it is exactly the opposite ! The stationarity property is way stronger in images and local correlation are largely more present than in spectrograms. You take the example of formants, which is actually the strongest local correlation that you could find, and even then vocal formants still follow an harmonic structure (hence global and cyclical correlation across frequencies). Any harmonic or noisy sound is a counter-example to this local correlation property.
- You talk about the high redundancy in the spectrogram (l. 118), and I think that analysing this aspect would make this paper stronger. How much it impacts the masked learning depending on the sound nature ?

**Limitations:**

-

**Strengths And Weaknesses:**

Overall, the use of MAE for audio is an interesting avenue of research, as they perfectly fit to the self-supervised approaches. The experiments appear sound and are numerous, which is one of the major strength of the paper.

However, the two main weaknesses of this paper is that the originality and amount of contributions is quite low (as the author state themselves it is a "conceptually simple extension of MAE to audio"). Although the authors study several properties of the model and input data, it feels more like an engineering paper and I would recommend to resubmit this paper to a more applied conference. Furthermore, the paper is sometimes not very well written with several long and hard to follow sentences. The same applies to the related work section, which is a laundry list of citations, without clearly delineating the advantages or flaws of past methods.

---

> ### Author Response · Authors · 2022-08-02
> **Response to Reviewer NJ9B (2/2)**
>
> **Q3** Local correlation in image/sound?
>
> We are grateful for reviewer NJ9B’s insight. We are aware that local correlation is also important for images when considering at the patch/sub-patch level (e.g. textures, continuity across patch boundaries). As we fully agree with reviewer NJ9B’s perspective that images can be very local; here is why we thought images could also be less local at a different level. We were advocating this in the paper and want to clarify why we think so. We observe that image patch distribution/correlation at the broader/extended level (e.g. whole image) plays a less significant role for determining image semantics (e.g. the size/position of a bird in an image may not be important; a "sky patch" could be anywhere on the upper part of an image).
>
> We consider local correlations are also important in spectrograms. At the contiguous/sub-patch level, human hearing perceptions are sensitive to defect/discontinuity in spectrograms, especially in 3-5KHz [60]. At the broader/extended level, a sound's pattern, patch distribution, and exact position in the spectrogram directly affects how it sounds and affects its semantics. To this end, we design Audio-MAE to better address local correlation of spectrogram patches for self-supervised learning from audio.
>
> In Audio-MAE, we promote a more locally-focused reconstruction task as the pretext task, where distant (in time and frequency) spectrogram context are less influential for reconstruction. As an intuitive example, along the time axis, spectrogram patches of speech are locally correlated and are less related to distant (in time) patches. Along the frequency axis, patches of frictional sounds in consonants (4-8KHz) or harmonics in formants (<2KHz) are locally correlated across several neighboring patches and are less related on distant (in frequency) patches. We achieve this by incorporating shifted local attention windows. In our experiment this mechanism qualitatively (Fig.5) and quantitatively (Table2) yields improved results.
>
> Thank you for pointing out strong local correlation in images and we will revise l.57 to make it more clear in the paper.
>
>
> **Q4** Masking analysis of the redundancy in sound nature?
>
> Thanks for the insightful suggestion. We conducted a new experiment to analyze how masking affects classification on different sound types. In the following table, we vary the masking ratio in self-supervised pre-training and show the average precision (AP) for different sound types after fine-tuning on AudioSet.
>
> |mask rate|speech|music|aircraft|env. noise|
> |-|-|-|-|-|
> |30%|82.1|77.1|55.8|49.4|
> |50%|82.9|78.6|56.1|49.6|
> |70%|83.3|80.5|55.9|49.7|
>
> Interestingly, sound types with comparably more harmonics or redundancy (e.g. music, +2.4) benefited more from increased masking ratio in Audio-MAE, while noise-like/high-entropy classes (e.g. env. noise, +0.3) do not. We will include more per-class analysis in the appendix to gain more insights for masking in Audio-MAE. This experimental insight opens up a new hypothesis that we would like to explore: Masking ratios could be adaptive w.r.t. the redundancy of the data (e.g. music shows a clear gain for higher masking ratios). We thank the reviewer for this suggestion.
>
>
> We hope our responses have covered and addressed your concerns. We are available and open to address any outstanding issues. Thank you for your insightful comments.

---

> > ### Comment · Area_Chair_RFB8 · 2022-08-10
> > **Answer to rebuttal**
> >
> > Dear reviewer, could you please address the rebuttal by the authors? The reviewer-AC discussion period starts tomorrow and comments will be closed to authors so they will not be able to address additional comments.

---

> ### Author Response · Authors · 2022-08-02
> **Response to Reviewer NJ9B (1/2)**
>
> Thank you for reviewing our work. We will extend related work with the additional page in the final version. For your questions:
>
> **Q1** “Conceptually simple" approach?
>
> Please note that conceptually simple works such as seq2seq [r3] have a long history of acceptance at top venues such as NeurIPS. We hope that simplicity is a strength, facilitating wide adoption within the community. Our Audio-MAE is a simple-yet-effective approach for self-supervised learning from audio. Further, instead of naively consuming spectrograms, our model also addresses the unique properties of sound which leads to improved performance. More broadly, we think that novelty can include technical innovation (e.g. our shifted local-attention decoders), as well as novel insights, ablations, and qualitative observations (e.g. our audible results). We are thankful that many of these properties are recognized by the reviewers.
>
>
> **Q2** "Unstructured" masking over rigid spectrogram patches?
>
> This is a great question. For unstructured/structured masking, we mean applying Bernoulli dropout/masking to the transformer token-level sequence (unstructured) or explicitly masking tokens along time/frequency axes (structured). The basic unit for masking is one token, that corresponds to the embedding of a 16x16 ‘pixel’ patch in the input spectrogram. We will revise this in l.106 to make it more clear.
>
> Following MAE, we use 16x16 patches as the smallest units and operate over their embeddings. This 16x16 patch setup is adopted by all the other transformer-based baselines in Table 2 as well (e.g. AST[10], SS-AST[18], MBT[11]). Compared to 16x16 patches, the major concern of using 1x1 “pixels” is the computation cost. Even with 80% masking, the input sequence length $N$ to the Transformer will be 102 with 16x16 patches and 26214 with 1x1 “pixels”, where the latter is too expensive (complexity is proportional to $O(N^2))$.
>
> We investigated various patch size configurations in Table 1a. Compared to 16x16 size (row1 47.3 mAP on AS-2M), finer and overlapping patches with 10x10 stride resulted in ~2.7x increased computation cost but results (row2 47.3) are similar.
>
> Following the reviewers’ suggestion, we additionally conduct an experiment with more fine-grained 8x8 patches, which increases the computational cost by ~3.6x, to 175G FLOPs, and is the finest patch encoding we can fit into V100 GPU memory. The result for our Audio-MAE converged to 47.3 and we will include this in Table1a of the main paper. We thank the reviewer for this suggestion.
>
> [r3] I. Sutskever et al. “Sequence to sequence learning with neural networks.” NIPS 2014.

---

### Official Review · Reviewer_JVUa · 2022-07-11

**Rating:** 8
**Confidence:** 4
**Soundness:** 3 good
**Presentation:** 4 excellent
**Contribution:** 4 excellent

**Summary:**

The paper proposes a method for self-supervised learning for audio classification. It is based on the prior work MAE which does self-supervised learning on images. The paper under review adapted the MAE model to audio by uses the spectrogram as an image. The decoder was changed to use the local self-attention as the audio tens to have more local correlations.

Then, the paper investigates the proposed architecture on several audio classification datasets. The paper conducts ablation studies on several aspects of the model: the masking strategies (finding that the unstructured masking works best for pre-training, and time+frequency masking works best for fine-tuning), the patch size and stride (finding that the overlap is not necessary), the encoder size, the decoder local or global attention, the length of the pre-training, and using the out-of-domain data for pre-training.

Finally, the paper compares the results to the previously published works. The proposed model outperforms all the previous in-domain models. Then, the in-domain proposed model matches or outperforms some of the out-of-domain methods on some datasets, while it is worse on other datasets.

**Questions:**

The paper argues in 4.4 that the out-of-domain training is not beneficial or even harmful. Nevertheless, the Table 2 shows that other methods benefit from the out-of-domain training. This needs some clarification.

**Strengths And Weaknesses:**

# Strengths

The proposed model is sound. Applying and testing the CV models on audio is important and poses its own challenges which were overcome in this publication.

This paper conducts a thorough experimentation on multiple datasets with many ablations. The experimental section contains a lot of information and very interesting to read.

In general, the paper is very well written and easy to follow. I was able to all the logical steps and transitions. The amount of details is not overwhelming, but provides all the necessary information.

# Weaknesses

While I find it logical that pre-training on image data should not help much for audio training, the previous publications find it useful. This paper concludes that the out-of-domain data is not beneficial. This looks like an important area to address (see the question below).

---

> ### Author Response · Authors · 2022-08-02
> **Response**
>
> We thank the reviewer for pointing out one of our important findings in Table 1h that compared to audio-only pre-training, the additional out-of-domain ImageNet pre-training (IN-PT) relatively degrades fine-tuning performance for audio tasks (47.3 $\rightarrow$ 46.9 mAP on AS-2M).
>
> **Q1** Why does ImageNet-PT no longer help?
>
> Compared to randomly initializing audio models from scratch, previous work on audio models (e.g. AST[10], MBT[11]) leverage models supervisedly pre-trained on ImageNet-1K or ImageNet-21K. The underlying assumption is that the patterns of visual objects to some extent resemble the patterns of spectrograms, and the knowledge to classify IN classes (dogs, cats) can be transferred to audio events (music, speech). Empirically they showed that using out-of-domain IN-PT resulted in faster convergence and better performance compared to random (“from-scratch”) initialization after fine-tuning for audio tasks. Some audio-only pre-training models have been proposed (e.g. Conformer[37], SS-AST[18]) to replace IN-PT yet still lag in performance, especially in the challenging AudioSet tasks.
>
> Our Audio-MAE is the first audio-only pre-training work that achieves state-of-the-art performance on AudioSet tasks. In this work we systematically study and compare the impact of using out-of-domain ImageNet for pretraining (i.e., self-supervised IN-PT with IN data and supervised IN-PT using IN labels). We show that the audio-only setup is sufficient to achieve the best performance (47.3 mAP) in Table 1h, outperforming other baseline with IN-PT in Table 2. Using only audio information, Audio-MAE does not rely on cross-modal knowledge transfer, i.e., pattern and class similarity between visual objects and audio spectrograms, as in prior work. In fact, incorporating self-supervised IN-PT MAE degrades performance (47.3 $\rightarrow$ 47.1) and incorporating ImageNet labels in finetuned MAE representations degrades accuracy further (47.3 $\rightarrow$ 46.9 in Table 1h).
>
> In short, our results suggest that image-to-audio transfer is suboptimal compared to using true audio spectrograms for pre-training, under the Audio-MAE setup. In previous works, initialization with IN-PT was a useful approach compared to from-scratch initialization (i.e., IN-PT > AS-PT > from-scratch). This is not helpful for Audio-MAE, where audio-only training is sufficient and achieves better performance (i.e., AS-PT > IN-PT > from-scratch).
>
> We hope our responses have covered and addressed your concerns. We are available and open to address any outstanding issues. Thank you for your insightful comments.

---

> > ### Comment · Area_Chair_RFB8 · 2022-08-10
> > **Answer to rebuttal**
> >
> > Dear reviewer, could you please address the rebuttal by the authors? The reviewer-AC discussion period starts tomorrow and comments will be closed to authors so they will not be able to address additional comments.

---

### Official Review · Reviewer_THYp · 2022-07-12

**Rating:** 4
**Confidence:** 4
**Soundness:** 2 fair
**Presentation:** 4 excellent
**Contribution:** 1 poor

**Summary:**

The author apply Masked Autoencoders from the vision domain to learn representations from audio spectrograms in a self-supervised fashion. The learned representaions can be fine-tuned to achieve competetive performance on different tasks. The learned representations can be fine-tuned for competitive performance on various tasks. The authors propose using local attention windows.

**Questions:**

I've listed the questions in the weaknesses section.

**Limitations:**

- The authors were up front about the similarity of their work to [42] [38]
- The authors do not justify the claims about ImageNet transfer learning.
- The large computational requirements compared to ImageNet transfer learning, with little improvments.

**Strengths And Weaknesses:**

Strengths:
- Clear and well-written presentation.
- Achieve SOTA on Audioset in a semi-supervised setup
- Doesn't relay on Imagenet and knowledge transfer from the vision domain.
- Extensive experiments on Audioset.

Weaknesses:
1. Low originality: the authors use MAE on spectrograms. I propose emphasizing the contributions of this work more clearly in the introduction (for example, adapting the local attention windows). Similar concurrent work  [42] [38] (as the authors point out clearly) exists. In particular, MaskSpec[42] is very close to the proposed work.  In light of this, what are the contibutions of your work(question to the authors)?

2. The authors argue repeatedly against knowledge transfer from Imagenet. However, the claims against are not quite justified, it's not clear what the drawbacks of imagenet are (question to the authors)?. For example, in line 32, what do you mean by label bias in this situation?

3. The huge computational complexity, even for fine tuning, especially compared to models pre-trained on Imagenet.
    - For finetuning the supervised methods on Audioset: AST[10] can be trained on 4  (titan RTX) GPUS in 1 week. PaST[28]  (also uses masking with Patchout) in 25-50 hours on a single 2080ti.
    - In SSL: MaskSpec[42] takes 4 days on 8x  V100 for the pretraining phase. I think this can be close in compute requirements to what the authors report?
    -  How do you justify the huge computation complexity with the limited improvement (within the error bars) on Audioset? (question to the authors)
4.  The large computation complexity limits the usability of the models to extract representations (in my opinion).  (I suggest a large scale evaluation of the learned representaions on the tasks of the NeurIPS21 HEAR challange for example, https://hearbenchmark.com/ )

---

> ### Author Response · Authors · 2022-08-02
> **Response to Reviewer THYp**
>
> Thank you for reviewing our work and the constructive feedback.
>
> **Q1** Being upfront with concurrent work:
>
> Our original manuscript included MaskSpec[42] as concurrent work (uploaded to arXiv on 4/27, 3 weeks before NeurIPS deadline), to be upfront with any existing work. Please note that it should not be considered as prior work, since it is less than the 2-month period in  [NeurIPS-FAQ](https://neurips.cc/Conferences/2022/PaperInformation/NeurIPS-FAQ). Similarly for [38] which was posted 3/31 on arXiv, less than 2 months before NeurIPS deadline. We hope our submission will not be penalized by the concurrent work. We have added [42] as concurrent work into our related work and comparisons to be upfront with it.
>
> Following the reviewer’s suggestion, we emphasize some contributions:
> 1) Audio-MAE is different (e.g., applying local attention with shifted windows).
> 2) Audio-MAE's performance is much better (e.g., 37.1 vs 32.3 mAP in AS-20K, 98.3 vs 97.7 acc on SPC).
> 3) Our  work systematically investigates the impact of incorporating self-supervised and supervised ImageNet transfer for audio pre-training.
> 4) Our work presents comprehensive qualitative results and audible insights (in Fig. 6 and Fig. 1 in the supplementary) that we hope are useful for the community.
>
> **Q2** Cons of transferring from ImageNet
>
> The main drawback of ImageNet transfer is the reduced accuracy of our approach, shown in the experiments in Table 1h and discussed in l.284-297.
>
> Compared to audio-only pre-training on AudioSet, transferring from models pre-trained on out-of-domain ImageNet data degrades accuracy (row4, 47.3 $\rightarrow$ 47.1(IN SSL)). Transferring from models trained with ImageNet labels further degrades accuracy (row5, 47.3 $\rightarrow$ 46.9 (IN-SSL+IN-SL). We term this “label bias” in l.32, where we mean performance degradation due to domain shift and heterogeneity between visual and audio classes.
>
> We are not against ImageNet pre-training per se since ImageNet pre-training could achieve faster convergence as shown in [r1-r2]. But our results indicate that, given our setup in Audio-MAE, transferring models trained with ImageNet data (visual objects) and labels (cats, dogs, etc.) to audio models does not lead to improvements compared to directly learning from large-scale audio data alone (e.g. AudioSet). Nevertheless, we agree that there are some benefits of ImageNet pre-training, e.g. many models are available.
>
> **Q3** Computation complexity is worse in pre-training and fine-tuning/feature extraction?
>
> For comparing runtime across different systems and infrastructures, there are many factors that could impact the comparison (e.g. number of training epochs, length of each epoch, regularization, data fetching, performing video decoding or not, etc). For comparing to other works, we measure complexity with FLOPs as these are a hardware-independent measure of complexity, specified in Table 1a.
>
> In pre-training, Audio-MAE is far more efficient than the main baseline SS-AST [10] since 80% of spectrogram patches are dropped before encoding in Audio-MAE. Audio-MAE's FLOPs and parameters are close to concurrent MaskSpec [42] (both with ViT-B or ViT-S as backbone) & Audio-MAE is more accurate, e.g., in AS-20K, 37.1 vs 32.3 mAP with ViT-B and 32.1 vs 28.9 with ViT-S.
>
> Fine-tuning or feature extraction is based on a VIT-B transformer where the FLOPs (48.6G) and parameters (86M) are identical to other transformer baselines. Please note that, given similar or better (we adopt masking at fine-tung hence sequence length is shorter than baselines) computation complexity, Audio-MAE achieves state-of-the-art performance in all the tasks. For example, acc on SPC: Audio-MAE: 98.3; SS-AST: 98.0; MaskSpc: 97.7; AST: 98.1.
>
> We thank the reviewer for pointing out the importance of comparing computational complexities and will add them to the final paper.
>
>
> **Q4**  The large computational requirements compared to ImageNet transfer learning, with little improvements?
>
> For a fair comparison of transferring/pre-training costs, we think that ImageNet (IN) pre-training cost should also be included. For example, Image-MAE trains on 1.2M IN images for 1600 epochs, while Audio-MAE trains on 2M audio for 32 epochs. Consequently, if counting the full training cost, Audio-MAE-PT is more effective and achieves better performance, e.g. 47.3 mAP (AS-SSL+AS-FT) vs 45.4 (IN-SSL+AS-FT).
>
> Further, the proposed Audio-MAE (e.g., 37.1 mAP on AS-20K) significantly outperforms other models with IN-PT (e.g. AST (34.7), MBT (31.3)) or AS-PT (e.g. SS-AST (31.0), MaskSpec(32.3)). We nevertheless agree that ImageNet pre-training has its own merits as one can simply use off-the-shelf models provided by the community. We will make this clear in the paper.
>
> [r1] K. He et al, “Rethinking ImageNet pretraining,” CVPR 2019
>
> [r2] J. Li et al, “Audio AudioTagging Done Right,” arXiv.2203.13448
>
> Thank you and we are available to address any outstanding issues.

---

> > ### Comment · Area_Chair_RFB8 · 2022-08-10
> > **Answer to author rebuttal**
> >
> > Dear reviewer, could you please address the rebuttal by the authors? The reviewer-AC discussion period starts tomorrow and comments will be closed to authors so they will not be able to address additional comments.

---

### Official Review · Reviewer_v3sn · 2022-07-13

**Rating:** 8
**Confidence:** 4
**Soundness:** 4 excellent
**Presentation:** 4 excellent
**Contribution:** 3 good

**Summary:**

This paper extends earlier work on masked autoencoders (MAE) with an application on self-supervised learning paradigm for audio processing. The main subject that is subjected to is mel spectrogram. Both encoder and decoder acquires transformer-based architecture and local attention is deployed at decoder side, out-performing solely global attention. Such network takes advantages of the nature of spectrogram, which encodes correlation on local time and frequency bands.

**Questions:**

1. The dataset covered in the work and related tasks are not that common for speech processing research community. Could you please address why you used AudioSet to train the model?

2. For random masking, is ther any specific technique involved for generating the masks and define the default mask ratios? From earlier works? Or just pretty random thing we cannot tell?

**Limitations:**

There are two main limitations in this work which are not addressed:
1. Acquisition of mel spectrogram only. It would be good if we try on various types of spectrograms.
2. Since VoxCeleb is involved - a scoring framework might be beneficial for verification experiments.

**Strengths And Weaknesses:**

## Strengths
1. The improvement from the conventional MAE encoder is clearly explained and simple to grasp and re-produced, with detailed ablation analysis on receiptive field.
2. (Personal preference) The idea itself is closely related to the nature of spectrogram, instead of from the purely statistical perspective. Although the idea of regarding spectrogram as image patches is not new, this work put more emphasis on its relation with audio nature.

## Weaknesses
1. There are minor typos and grammatical mistakes in the paper. Please do proof-reading for the camera-ready version.
2. Mel spectrogram has its own limitations in information sparsity (especially for higher frequency regions). Therefore, raw or linear spectrogram might be better choices for this, at least as an additional ablation study.
3. VoxCeleb experiments. I am not 100% sure about how general ML community regard VoxCeleb, but officially it was proposed [1] as a verification task. Identification, in that case, can be regarded as an open-set verification.
4.  Probably some comparison with supervised models can be useful. But this is trivial.
5. There are some explanation in section I and II lacks justification or pointers to earlier studies, especially on explanining the nature of spectrogram's relation to speech cues - all things there makes sense to me, but needs some references.

---

> ### Author Response · Authors · 2022-08-02
> **Response to Reviewer v3sn**
>
> We thank you for your time and effort to review our manuscript, and appreciate the positive feedback!  We will include additional references and improve the writing quality in the final version. Regarding your questions:
>
> **Q1** Why not use linear spectrograms?
>
> Thanks for the suggestion. We have experimented with linear spectrograms and found that Mel spectrograms yield better performance (e.g. 37.3 vs 36.8 mAP on AS-20K). Presumably this is because the Mel filter banks align better with human hearing perception which in turn facilitates machine perception. Further, Mel spectrograms are slightly more efficient in computation (25ms window. 64x8 (Mel) vs 64x13 (linear) spectrogram tokens (smaller is more efficient) for 10-sec audio under 16K sampling rate).
>
>
> **Q2** Comparison with supervised models?
>
> This is a good suggestion and we can show more comparisons in the final version. For now, we compare Audio-MAE to other models with ImageNet supervised pre-training (IN-PT) in the bottom group of Table 2. Audio-MAE significantly outperforms these models, e.g., 37.1 mAP (Audio-MAE) vs 34.7 (AST[10]) and 31.3 (MBT[11]) on AS-20K. In Table 1h, we also ablate Audio-MAE with its own variants initialized with supervised ImageNet pretraining where we show that this is not beneficial over self-supervised audio-only pre-training for our setup.
>
>
> **Q3** Why choose AudioSet for pre-training?
>
> We use AudioSet since: 1) It is a large and diverse audio collection. 2) We would like to set a fair comparison with the baselines. In more detail:
>
> Firstly, AudioSet is large (2M audio clips) and covers a wide range of sound types (~40% of audio clips are speech and the rest are event sounds or music). We leverage AudioSet for pre-training with a hope that the pre-trained model can generalize to various downstream tasks (e.g., audio and speech classification tasks). Secondly, AudioSet is the standard benchmark for audio event detection and studied by other baselines listed in Table 2. We would like to set a fair comparison with them.
>
>
> **Q4** Random masking technique?
>
> The masks are generated randomly as in image-MAE[1], on the token level.
>
> - For unstructured masking, we randomly sample a $p$ portion of all tokens (each token corresponds to one spectrogram patch) and mask the sampled ones (Fig. 2b). For every token, this can be regarded as a Bernoulli process with a ratio *p* being masked.
>
> - For structured masking, we randomly sample a $p$ portion of time frame indexes or frequency bands indexes then mask them. For time frames (e.g. the highlighted vertical stripes in Fig. 2c) or frequency bands (e.g. the highlighted horizontal stripes in Fig. 2d), this can be regarded as Bernoulli processes with a ratio $p$ being masked.
>
> We experimented with different masking types and ratios in Fig. 4 to find the best strategy, and will make this description more clear in the paper.
>
> Finally, thank you for the suggestion regarding Voxceleb, we will consider using the SUPERB[53] platform to test out other Voxceleb setups and other speech tasks. We will further add justification and references to earlier studies, e.g. on the nature of spectrograms, as suggested. Thank you. We hope our response have answered the reviewer's questions, but if not we are available and open to addressing any outstanding ones.

---

> > ### Comment · Area_Chair_RFB8 · 2022-08-10
> > **Reviewers, please address the rebuttal by the authors**
> >
> > Dear reviewer, could you please address the rebuttal by the authors? The reviewer-AC discussion period starts tomorrow and comments will be closed to authors so they will not be able to address additional comments.

---

> > ### Comment · Reviewer_v3sn · 2022-08-10
> > **Response of the comment**
> >
> > Thanks a lot for the explanation! Yes so far I have no problem with the comments from the authors except one thing: in terms of VoxCeleb, it would be good to follow the standard VoxSRC protocol from recent works on speaker verification. Speaker identification in my personal honest opinion is not a good metric for VoxCeleb due to its trial design.

---

> > > ### Author Response · Authors · 2022-08-10
> > > **Thank you for your response!**
> > >
> > > Thank you very much for the response! We are glad that most of your concerns have been properly addressed. We will experiment and include the speaker verification experiment following the suggested protocol in VoxSRC.

---

### Meta-Review · Area_Chair_RFB8 · 2022-08-25

**Recommendation:** Accept
**Confidence:** Less certain

**Metareview:**

The paper has two strong accepts and two borderline reject reviews. However, as one of the reviewers did not engage with the authors post-rebuttal, I had to interpret the authors' response to the reviewer's concerns, and they seem to properly address them (even including a new experiment into the paper). The work seems to have been executed concurrently with other similar approaches, and while not entirely novel, the paper seems to include in-depth experiments and a discussion that can be beneficial to the research community.

**Award:**

No

---

### Decision · Program_Chairs · 2022-09-14

Accept